# Unsupervised Learning of Dense Visual Representations

**Pedro O. Pinheiro**[1]**, Amjad Almahairi, Ryan Y. Benmalek**[2]**,**
**Florian Golemo**[13]**, Aaron Courville**[34]
[1]Element AI, [2]Cornell University, [3]Mila, Université de Montréal,
[4]CIFAR Fellow

## Abstract

Contrastive self-supervised learning has emerged as a promising approach to unsupervised visual representation learning. In general, these methods learn *global* (image-level) representations that are invariant to different views (*i.e.*, compositions of data augmentation) of the same image. However, many visual understanding tasks require *dense* (pixel-level) representations. In this paper, we propose View-Agnostic Dense Representation (VADeR) for unsupervised learning of dense representations. VADeR learns pixelwise representations by forcing local features to remain constant over different viewing conditions. Specifically, this is achieved through pixel-level contrastive learning: matching features (that is, features that describes the same location of the scene on different views) should be close in an embedding space, while non-matching features should be apart. VADeR provides a natural representation for dense prediction tasks and transfers well to downstream tasks. Our method outperforms ImageNet supervised pretraining (and strong unsupervised baselines) in multiple dense prediction tasks.

## 1   Introduction

Since the introduction of large-scale visual datasets like ImageNet [9], most success in computer vision has been primarily driven by supervised learning. Unfortunately, most successful approaches require large amounts of labeled data, making them expensive to scale. In order to take advantage of the huge amounts of unlabeled data and break this bottleneck, unsupervised and semi-supervised learning methods have been proposed.

Recently, self-supervised methods based on *contrastive learning* [23] have shown promising results on computer vision problems [73, 55, 30, 80, 29, 63, 1, 26, 49, 5]. Contrastive approaches learn a similarity function between views of images—bringing views of the same image closer in a representation space, while pushing views of other images apart. The definition of a view varies from method to method but views are typically drawn from the set of data augmentation procedures commonly used in computer vision.

Current contrastive self-supervision methods have one thing in common: similarity scores are computed between *global representations* of the views (usually by a global pooling operation on the final convolutional layer). See Figure 1a. Global representations are efficient to compute but provide low-resolution features that are invariant to pixel-level variations. This might be sufficient for few tasks like image classification, but are not enough for dense prediction tasks.

*Dense representations*, contrary to their global counterpart, yield encodings at pixel level. They provide a natural way to leverage intrinsic spatial structure in visual perception [70] during training (*e.g.*, nearby pixels tend to share similar appearances, object boundaries have strong gradients, object-centric properties). Moreover, many—arguably, most—visual understanding tasks rely on structured, dense

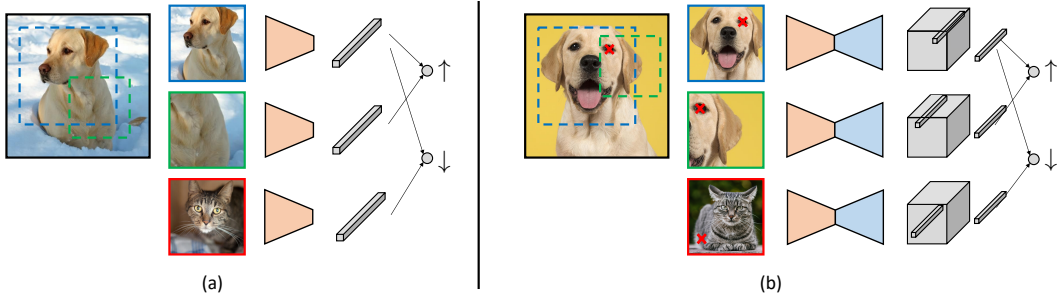

Figure 1: Contrastive methods learn representations by first generating two (correlated) views of a same scene and then bringing those views closer in an embedding space, while pushing views of other scenes apart. (a) Current methods compute similarity on global, image-level representations by applying a pooling operation on the output of the feature encoder. (b) VADeR, on the other hand, utilizes an encoder-decoder architecture and computes similarity on pixel-level representations. VADeR uses known pixel correspondences, derived from the view generation process, to match local features.

representations (*e.g.* pixel-level segmentation, depth prediction, optical-flow prediction, keypoint detection, visual correspondence).

In this paper, we propose a method for unsupervised learning of dense representations. The key idea is to leverage *perceptual constancy* [56]—the idea that local visual representations should remain constant over different viewing conditions—as a supervisory signal to train a neural network. Perceptual constancy is ubiquitous in visual perception and provides a way to represent the world in terms of invariants of optical structure [18, 69]. The local representations of a scene (*e.g.*, an eye of a dog) should remain invariant with respect to which viewpoint the scene is been observed.

Our method—*View-Agnostic Dense Representation* (VADeR)—imposes perceptual constancy by contrasting local representations. Here, representations of matching features (that is, features that describe the same location of a scene on different views) should be close in an embedding space, while non-matching features should be apart. Our method leverages known pixel correspondences, derived from the view generation process, to find matching features in each pair of views. VADeR can be seen as a generalization of previous contrastive self-supervised methods in the sense that it learns dense (*i.e.*, per-pixel) features instead of global ones. Figure 1b describes our general approach and compares it to common contrastive approaches.

VADeR provides a natural representation for dense prediction tasks. We evaluate its performance by seeing how the learned features can be transferred to downstream tasks, either as feature extractor or used for fine-tuning. We show that (unsupervised) contrastive learning of dense representation are more effective than its global counterparts in many visual understanding tasks (instance and semantic segmentation, object detection, keypoint detection, correspondence and depth prediction). Perhaps more interestingly, VADeR unsupervised pretraining outperforms ImageNet supervised pretraining at different tasks.

## 2 Related Work

**Self-supervised learning.**    Self-supervised learning is a form of unsupervised learning that leverages the intrinsic structure of data as supervisory signal for training. It consists of formulating a predictive pretext task and learning features in a way similar to supervised learning. A vast range of pretext tasks have been recently proposed. For instance, [65, 57, 78, 2] corrupt input images with noise and train a neural network to reconstruct input pixels. Adversarial training [20] can also be used for unsupervised representation learning [16, 12, 13]. Other approaches rely on heuristics to design the pretext task, *e.g.*, image colorization [77], relative patch prediction [10], solving jigsaw puzzle [52], clustering [3] or rotation prediction [19].

More recently, methods based on contrastive learning [23] have shown promising results for learning unsupervised visual representations. In this family of approaches, the pretext task is to distinguish between compatible (same instance) and incompatible (different instances) views of images, although the definition of view changes from method to method. Contrastive predictive coding (CPC) [55, 29] has been applied to sequential data (space and time) and define a view as been either past or future

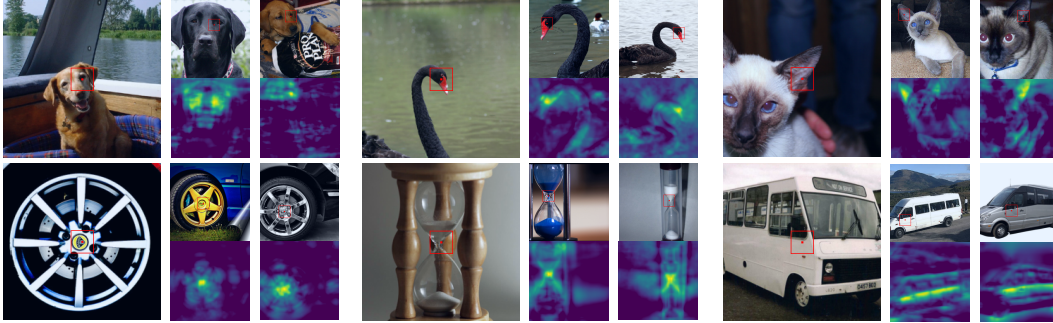

Figure 2: Pixel-level retrieval results. On left (large), we show the query pixel (center of image). On the right (small), we show the two images with the closest pixels (in embedding space) to the query pixel and the similarity map between the query pixel embedding and all pixel embeddings on the image.

observations. DeepInfomax [30, 1] train a network by contrasting between global (entire image) and local (a patch) features from same image.

Other methods [11, 73] propose a non-parametric version of [15], where the network is trained to discriminate every instance in the dataset. In particular, InstDisc [73] rely on noise-contrastive estimation [22] to contrast between views (composition of image transformations) of the same instance and views of distinct instances. The authors propose a memory bank to store the features of every instance to efficiently consider large number of negative samples (views from different images) during training. Many works follow this approach and use a memory bank to sample negative samples, either considering the same definition of view [80, 74, 26] or not [63, 49]. SimCLR [5] also consider a view as a stochastic composition of image transformations, but do not use memory bank, instead improving performance by using larger batches. Our method can be seen as an adaptation of these methods, where we learn pixel-level, instead of image-level, features that are invariant to views.

**Dense visual representations.** Hand-crafted features like SIFT [46] and HOG [8] has been heavily used in problems involving dense correspondence [43, 68, 35, 32]. Long et al. [45] show that a deep features trained for classification on large dataset can find correspondence between object instances, performing on par with hand-crafted methods like SIFT Flow [43]. This motivated many research on training deep networks to learn dense features [25, 75, 76, 6, 34, 53, 60, 64, 54], but these methods usually require labeled data and have very specialized architectures. In particular, [34] leverage correspondence flow (generated by image transformation) to tackle the problem of keypoint matching.

Recently, unsupervised/self-supervised methods that learn structured representations have been proposed. In [79], the authors propose a method that learn dense correspondence through 3D-guided cycle-consistency. Some work learn features by exploiting temporal signal, *e.g.*, by learning optical flow [14], coloring future frames [66], optical-flow similarity [47], contrastive predictive coding [24] or temporal cycle-consistency [67, 40]. Others, propose self-supervised methods to learn structured representations that encode keypoints [62, 33, 61, 48] or parts [31]. Our approach differs from previous methods in terms of data used, loss functions and general high-level objectives. VADeR learns general pixel-level representations that can be applied in different downstream tasks. Moreover, our method does not require specific data (such as videos or segmented images of faces) for training.

## 3 Method

Our approach, VADeR, learns a latent space by forcing representations of pixels to be viewpoint-agnostic. This invariance is imposed during training by maximizing per-pixel similarity between different views of a same scene via a contrastive loss. Conceptually, a good pixel-level embedding should map pixels that are semantically similar close to each other. Figure 2 shows selected results of VADeR learned representations. These qualitative examples hints that VADeR can, to a certain extend, cluster few high-level visual concepts (like eye of dog, beak of swan, ear of cat), independent of viewpoint and appearance and without any supervision. In the following, we first describe how we learn viewpoint-agnostic dense features. Then we describe our architecture and implementation details.

## 3.1 View-Agnostic Dense Representations

We represent a pixel $u$ in image $\mathbf{x} \in \mathcal{I} \subset \mathbb{R}^{3 \times h \times w}$ by the tuple $(\mathbf{x}, u)$. We follow current self-supervised contrastive learning literature [73, 26, 5] and define a *view* as a stochastic composition of data transformations applied to an image (and its pixels). Let $(v\mathbf{x}, vu)$ be the result of applying the view $v$ on $(\mathbf{x}, u)$. A view can modify both appearance (*e.g.* pixel-wise noise, change in illumination, saturation, blur, hue) and geometry (*e.g.* affine transform, homographies, non-parametric transformations) of images (and their pixels).

Let $f$ and $g$ be encoder-decoder convolutional networks that produce $d$-dimensional embedding for every pixel in the image, *i.e.*, $f, g : (\mathbf{x}, u) \mapsto z \in \mathbb{R}^d$. The parameters of $f$ and $g$ can be shared, partially shared or completely different. Our objective is to learn an embedding function that encodes $(\mathbf{x}, u)$ into a representation that is invariant w.r.t. any view $v_1, v_2 \in \mathcal{V}_u$ *containing the original pixel* $u$. That is, $f(v_1\mathbf{x}, v_1 u) = g(v_2\mathbf{x}, v_2 u)$ for every pixel $u$ and every viewpoint pairs $(v_1, v_2)$. To alleviate the notation, we write the pixel-level embeddings as $f_{v_1 u} = f(v_1\mathbf{x}, v_1 u)$, $g_{v_2 u} = g(v_2\mathbf{x}, v_2 u)$ and $g_{u'} = g(\mathbf{x}', u')$. Ideally, we would like to satisfy the following constraint:

$$c(f_{v_1 u}, g_{v_2 u}) > c(f_{v_1 u}, g_{u'}), \forall u, u', v_1, v_2, \tag{1}$$

where $u$ and $u'$ are different pixels, and $c(\cdot, \cdot)$ is a measure of compatibility between representations.

In practice, the constraint above is achieved through contrastive learning of pixelwise features. There are different instantiations of contrastive loss functions [39], *e.g.*, margin loss, log-loss, noise-contrastive estimation (NCE). Here, we adapt the NCE loss [22, 55] to contrast between pixelwise representations. Intuitively, NCE can be seen as a binary classification problem, where the objective is to distinguish between compatible views (different views of the *same* pixel) and incompatible ones (different views of *different* pixels).

For every pixel $(\mathbf{x}, u)$, we construct a set of $N$ random negative pixels $\mathcal{U}^- = \{(\mathbf{x}', u')\}$. The loss function for pixel $u$ and $\mathcal{U}^-$ can be written as:

$$\mathcal{L}(u, \mathcal{U}^-) = -\mathbb{E}_{(v_1, v_2) \sim \mathcal{V}_u} \left[ \log \frac{\exp(c(f_{v_1 u}, g_{v_2 u}))}{\exp(c(f_{v_1 u}, g_{v_2 u})) + \sum_{u' \in \mathcal{U}^-} \exp(c(f_{v_1 u}, g_{u'}))} \right]. \tag{2}$$

We consider the compatibility measure to be the temperature-calibrated cosine similarity, $c(\mathbf{x}_1, \mathbf{x}_2) = \frac{1}{\tau} \mathbf{x}_1^T \mathbf{x}_2 / \|\mathbf{x}_1\| \|\mathbf{x}_2\|$, where $\tau$ is the temperature parameter. By using a simple, non-parametric compatibility function, we place all the burden of representation learning on the network parameters.

This loss function forces representations of a pixel to be more compatible to other views of the same pixel than views of other pixels. The final loss consists on minimizing the empirical risk over every pixel on every image of the dataset.

## 3.2 Implementation Details

**Sampling views.** We consider view as been a composition of (i) appearance transformations (random Gaussian blur, color jitter and/or greyscale conversion) and (ii) geometric transformations (random crop followed by resize to $224 \times 224$). In this work, we use the same transformations as in [26].

Positive training pairs are generated by sampling pairs of views on each training image and making sure at least 32 pixels belong to both views (we tried different minimum number of matching pixels per pair, and did not notice any quantitative difference in performance). Pixel-level matching supervision comes for free: the two views induce a correspondence map between them. We use this correspondence map to find the matching features between the pair of views and consider each matching pair as a positive training sample.

The number of negative samples is particularly important if we consider pixelwise representations, as the number of pixels in modern datasets can easily achieve the order of hundreds of billions. In preliminary experiments, we try to use different pixels of the same image as negative examples but fail to make it work. We argue that using pixels from other images is more natural for negative samples and it fits well in the context of using a queue for negative samples. We use the recently proposed momentum contrast mechanism [26] to efficiently use a large number of negative samples during training. The key idea is to represent the negative samples as a queue of large dynamic dictionary that covers a representative set of negative samples and to model the encoder $g$ as a momentum-based

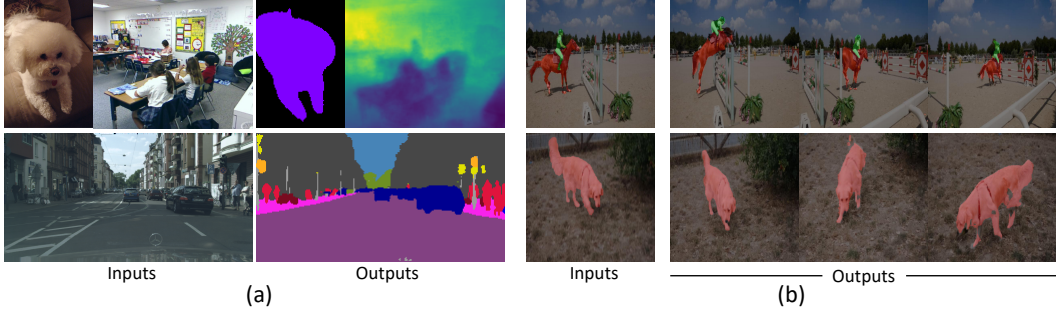

Figure 3: Example of task evaluated (shown are the output of VADeR in different settings). (a) Semantic segmentation and depth estimation can be seen as structured multi-class classification and regression problem, respectively. (b) Video instance segmentation, where the instance label is given for the first frame, and is propagated through the frames.

moving average of $f$. Following [26], we set the size of the dictionary to 65,536 and use a momentum of 0.999. We observe similar behavior w.r.t. the size of memory bank as reported in [26].

**Architecture.** We adopt the feature pyramid network (FPN) [41] as our encoder-decoder architecture. FPN adds a lightweight top-down path to a standard network (we use ResNet-50 [28]) and generates a pyramid of features (with four scales from $1/32$ to $1/4$ resolution) with dimension 256. Similar to the semantic segmentation branch of [36], we merge the information of FPN into a single dense output representation. At each resolution of the pyramid, we add a number of upsampling blocks so that each pyramid yield a feature map of dimension 128 and scale $1/4$ (*e.g.*, we add 3 upsampling blocks for res. $1/32$, 1 upsampling for $1/8$ res.). Each upsampling block consists of a $3 \times 3$ convolution, group norm [71], ReLU [50] and $2\times$ bilinear upsampling. Finally, we element-wise sum the pyramid representations and pass through a final $1 \times 1$ convolution. The final representation has dimension 128 and scale $1/4$. Since we use images of size $224 \times 224$ during training, the feature map generated by VADeR has dimension $128 \times 56 \times 56$.

**Training.** We train our model on the ImageNet-1K [9] train split, containing approximately 1.28M images. The weights of $f$ are optimized with stochastic gradient descent with weight decay of 0.0001 and momentum 0.9. We train using 4 GPUs with a batch size of 128 for about 6M iterations. We use a learning rate of $3e^{-7}$ and $3e^{-3}$ for the encoder and decoder, respectively. We multiply Equation 2 by a factor of 10, as we observed it provides more stable results when fine-tuning with very small amount of labeleld data. We set the temperature $\tau$ to 0.07.

Training samples are generated by sampling a random image and two views. We store the correspondence flow between the pixels that belongs to both view-transformed images. After forwarding each view through the networks, we apply the loss in Equation 2 considering 32 pairs of matching pixels as positive pairs (we chose randomly 32 pairs out of all matching pairs). The negative samples are elements from the MoCo queue. We then update the parameters of network $f$ with SGD and the weights of $g$ with moving average of the weights of $f$. Finally, we update the elements of the dynamic dictionary and repeat the training.

We opt to initialize the weights of VADeR's encoder with MoCo [26] to increase training speed. The weights of the decoder are initialized randomly. We consider MoCo[1] as our most important baseline for two reasons: (i) it is current state of the art in unsupervised visual representation and (ii) we start from MoCo initialization. We also compare our model with a ResNet-50 [28] pre-trained on ImageNet-1K with labels. MoCo contains the same capacity of ResNet-50 and VADeR contains around 2M extra parameters due to the decoder.

## 4 Experimental Results

An important objective of unsupervised learning is to learn features that are transferable to downstream tasks. We evaluate the quality of the features learned by VADeR on a variety of tasks ranging from

recognition to geometry (see Figure 3 for examples of tasks evaluated). We consider two transfer learning experimental protocols: (i) *feature extraction*, where we evaluate the quality of fixed learned features, and (ii) *fine-tuning*, where we use the learned features as weight initialization and the whole network is fine-tuned. For each experiment, we consider identical models and hyperparameters for VADeR and baselines, only changing the fixed features or the initialization for fine-tuning. For more details about datasets and implementations of downstream tasks, see supplementary material.

## 4.1 Feature Extraction Protocol

**Semantic segmentation and depth prediction.** We follow common practice of self-supervised learning [21, 37] and assess the quality of features on fixed image representations with a linear predictor. The linear classifier is a $1 \times 1$ convolutional layer that transform the features into logits (followed by a softmax) for per-pixel classification (semantic segmentation) or into a single value for per-pixel regression (depth prediction). We train the semantic segmentation tasks with cross entropy and the depth prediction with L1 loss. We test the frozen features in two datasets for semantic segmentation (PASCAL VOC12 [17] and Cityscapess [7]) and one for depth prediction (NYU-depth v2 [51]). In all datasets, we train the linear model on the provided train set[2] and evaluate on the validation set.

We compare VADeR with randomly initialized network (which serves as a lower bound), supervised ImageNet pretraining and MoCo (all with ResNet-50 architecture). Because baselines are trained for global representations, we need to adapt them to be more competitive for dense prediction tasks. We first remove the last average pooling layer so that the final representation has a resolution of $1/32$. Then, we reduce the effective stride to $1/4$ by replacing strided convolution with dilated ones, following the large field-of-view design in [4]. This way, the baselines produce a feature map with same resolution as VADeR.

|  | sem. seg. (mIoU) | | depth (RMSE) |
|---|---|---|---|
|  | VOC | CS | NYU-d v2 |
| random | 04.9 | 10.6 | 1.261 |
| sup. IN | 54.4 | **47.1** | 0.994 |
| MoCo | 43.0 | 32.3 | 1.136 |
| VADeR | **56.7** | 44.3 | **0.964** |

Table 1: Sem. seg. and depth prediction evaluated with fixed features.

Table 1 compares results (averaged over 5 trials) in standard mean intersection-over-union (mIoU) and root mean square error (RMSE). VADeR outperform MoCo in all tasks by a considerable margin. It also achieves better performance than supervised ImageNet pretraining in one semantic segmentation task and in depth prediction. This corroborates our intuition that explicitly learning structured representations provides advantages for pixel-level downstream tasks.

**Video instance segmentation.** We also use fixed representations to compute dense correspondence for instance segmentation propagation in videos. Given the instance segmentation mask of the first frame, the task is to propagate the masks to the the rest of the frames through nearest neighbours in embedding space. We evaluate directly on learned features, without any additional learning. We follow the testing protocol of [67] and report results in standard metrics, including region similarity $\mathcal{J}$ (IoU) and contour-based accuracy $\mathcal{F}$[3].

Table 2 shows results on DAVIS-2017 validation set [59], considering input resolution of $320 \times 320$ and $480 \times 480$. VADeR results are from a model trained on 2.4M iterations, as we observed that it performs slightly better on this problem (while performing equal or slightly worse on other tasks). Once again, we observe the advantage of explicitly modelling dense representations: VADeR surpass recent self-supervised methods [66, 67, 38] and achieve comparable results with current state of the art [40]. VADeR, contrary to competitive methods, achieve these results *without* using any video data nor specialized architecture. We note that MoCo alone is already a strong baseline, achieving similar performance to some specialized methods and close to supervised ImageNet pretrain.

## 4.2 Fine-tuning Protocol

In this section, we compare how good features are for fine-tuning on downstream tasks. All baselines have the same FPN architecture of VADeR (described in Section 3.2) and are trained with identical hyperparameters and data. Apart from weight initialization, everything else is kept identical—this allows for straight comparison of different initialization methods. We compare VADeR with two

| | $\mathcal{J}$ (Mean) | | $\mathcal{F}$ (Mean) | |
|---|---|---|---|---|
| | 320 | 480 | 320 | 480 |
| SIFT Flow [44] | 33.0 | - | 35.0 | - |
| Video Colorization [66] | 34.6 | - | 32.7 | - |
| TimeCycle [67] | 41.9 | 46.4 | 39.4 | 50.0 |
| CorrFlow [38] | - | 47.7 | - | 51.3 |
| UVC [40] | 52.0 | **56.8** | 52.6 | **59.5** |
| sup. IN | 50.3 | 53.2 | 49.2 | 56.2 |
| MoCo [26] | 42.3 | 51.4 | 40.5 | 54.7 |
| VADeR | **52.4** | 54.7 | **55.1** | 58.4 |

Table 2: Results on instance mask propagation on DAVIS-2017 [59] val. set, with input resolution of $320 \times 320$ and $480 \times 480$. Results are reported on region similarity $\mathcal{J}$ (IoU) and contour-based accuracy $\mathcal{F}$.

| view sampling | | sem. seg. mIoU | corr. $\mathcal{J}$ |
|---|---|---|---|
| unmatch | *fix.* | 32.4 | 9.8 |
| | *f.t.* | 74.1 | - |
| same view | *fix.* | 48.0 | 46.7 |
| | *f.t.* | 75.2 | - |
| diff. view | *fix.* | 56.7 | 52.4 |
| | *f.t.* | 75.4 | - |

Table 3: Results of VADeR considering different view sampling strategy for semantic segmentation (mIoU in VOC) and dense correspondence ($\mathcal{J}$ in DAVIS-2017). See text for details.

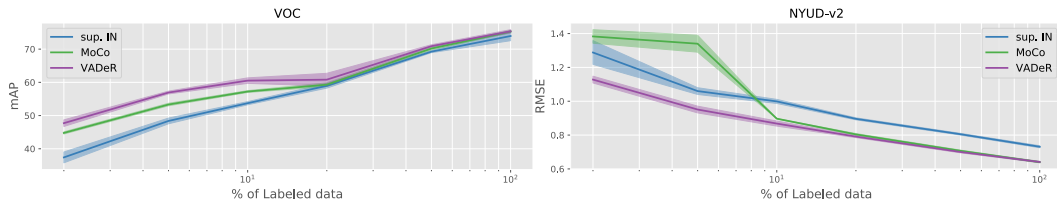

Figure 4: Results on semantic segmentation (VOC) and depth prediction (NYU-d v2) evaluated with fine-tuned features, considering different amount of labeled data. Evaluation is on val. set of each dataset. We show mean/std results over 5 trials in standard mean intersection-over-union (mIoU) and root mean square error (RMSE).

encoder initializations: supervised ImageNet pretraining and MoCo. VADeR, contrary to baselines, initializes both the FPN's encoder and decoder. In these experiments, we use the setup for fine-tuning proposed in [26]: the batch normalization layers are trained with Sync Batch-Norm [58] and we add batch-norm on all FPN layers.

**Segmentation and depth.** As in the previous experiment, we add an extra $1 \times 1$ convolutional layer on the output of the FPN architecture to perform either multiclass classification or regression. We use the same training and validation data as previous experiment (for training details, see supplementary material).

Figure 4 shows results of fine-tuning on semantic segmentation (in PASCAL VOC12) and in depth prediction (NYU-d v2), assuming different amount of labeled data (we consider 2, 5, 10, 20, 50 and 100% of dataset). When considering 100% of labeled images, VADeR achieves similar performance as MoCo (no statistical difference) and surpass supervised ImageNet pretraining. The benefits of VADeR, however, increase as the amount of labeled data on fine-tuning stages is reduced. This result corroborates current research that show that self-supervised learning methods achieve better performance than supervised pretraining when the number of labeled data is limited. Table 5 in supplementary material show results in tabular format.

**Object detection, instance segmentation and keypoint detection.** We use Mask R-CNN [27] with FPN backbone [41]. All methods are trained and evaluated on COCO [42] with the standard metrics. We use the implementation of `Detectron2` [72] for training and evaluation[4]. We train one model for object detection/segmentation and one for keypoint detection, using the default hyperparameters provided by [72] (chosen for ImageNet supervised pretraining). All models are trained on a controlled setting for around 12 epochs (schedule 1x).

Table 4 compares VADeR with baselines on the three tasks. MoCo is already a very strong baseline achieving performance similar to supervised ImageNet pretraining. We observe that VADeR consistently outperform MoCo (and the supervised baseline) on these experiments, showing advantages of learning dense representations contrary to global ones.

|  | $AP^{bb}$ | $AP_{50}^{bb}$ | $AP_{75}^{bb}$ | $AP^{mk}$ | $AP_{55}^{mk}$ | $AP_{75}^{mk}$ | $AP^{kp}$ | $AP_{50}^{kp}$ | $AP_{75}^{kp}$ |
|---|---|---|---|---|---|---|---|---|---|
| random | 31.0 | 49.5 | 33.2 | 28.5 | 46.8 | 30.4 | 65.4 | 85.8 | 70.8 |
| sup. IN | 39.0 | **59.8** | 42.5 | 35.4 | 56.6 | 37.8 | 65.3 | 87.0 | 71.0 |
| MoCo [26] | 38.9 | 59.4 | 42.5 | 35.4 | 56.3 | 37.8 | 65.7 | 86.8 | 71.7 |
| VADeR | **39.2** | 59.7 | **42.7** | **35.6** | **56.7** | **38.2** | **66.1** | **87.3** | **72.1** |

Table 4: Results on mask rcnn on object detection, instance segmentation and keypoint detection fine-tuned on COCO. We show results on val2017, averaged over 5 trials.

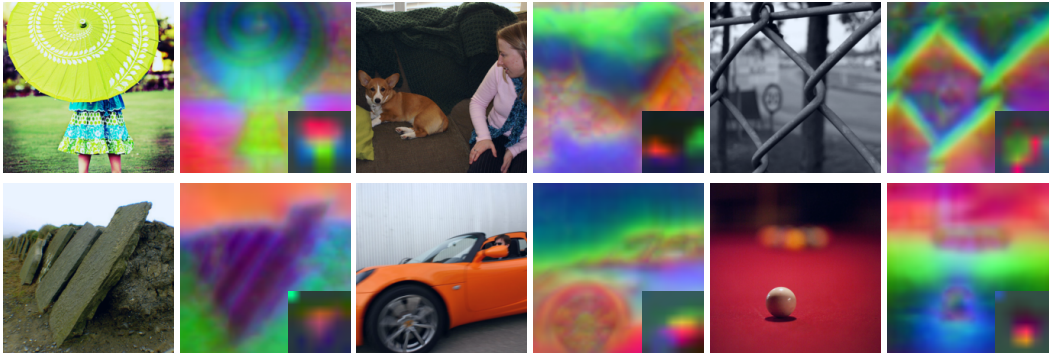

Figure 5: Visualizing the learned features. We show query images, VADeR (big) and MoCo (small, bottom-right) features. The features are projected into three dimensions with PCA and visualized as RGB (similar to [66]). Similar color implies similarity in feature space. Best viewed in color.

## 4.3 Ablation Studies

**Influence of views.** We analyze the importance of correctly matching pixels during training. Table 3 shows results considering three pixel-matching strategies. The first row ("unmatch") ignores the correspondence map between views of the same scene. It considers random pairs of pixels as positive pairs. Second and third rows utilize the correct correspondence map. The second row ("same view") always consider identical crop for each pair of views, while the last ("diff. view") consider different crops (with different location and scale). VADeR uses the third approach as default. Results are reported for semantic segmentation on Pascal VOC (in mIOU) and for dense correspondence on DAVIS-2017 (in region similarity $\mathcal{J}$). As expected, using random pixel matching between views is worse than using correct pixel pairing (row 1 vs. rows 2 and 3). We also note that performance in recognition task almost does not change when considering same or different crops between pairing views (row 2 vs. row 3). However, for correspondence, having difference crops per view provides a considerable advantage.

**Semantic grouping.** Figure 5 shows features learned by VADeR (big) and MoCo (small, bottom-right) projected to three dimensions (using PCA) and plotted as RGB. Similar colors implies that features are semantically similar. We observe qualitatively that *semantically meaningful grouping emerges from VADeR training, without any supervision.* This is an interesting fact that can potentially be useful in unsupervised or non-parametric segmentation problems.

## 5 Conclusion

We present VADeR—View-Agnostic Dense Representations—for unsupervised learning of dense representations. Our method learns representations through pixel-level contrastive learning. VADeR is trained by forcing representations of matching pixels (that is, features from different views describing same location) to be close in an embedding space, while non-matching features to be far apart. We leverage known pixel correspondences, derived from randomly generated views of a scene, to generate positive pairs. Qualitatively examples hints that VADeR features can discovery high-level visual concepts and that semantic grouping can emerge from training, without any label. We show that learning unsupervised dense representations are more efficient to downstream pixel-level tasks than their global counterparts. VADeR achieves positive results when compared to strong baselines in many structured prediction tasks, ranging from recognition to geometry. We believe that learning unsupervised dense representations can be useful for many structured problems involved transfer learning, as well as unsupervised or low-data regime problems.

# 6   Broader Impact

Our research falls under the category of advancing machine learning techniques for computer vision and scene understanding. We focus on improving image representations for dense prediction tasks, which subsumes a large array of fundamental vision tasks, such as image segmentation and object detection. While there are potentially many implications for using these applications, here we discuss two aspects. First, we highlight some social implications for image understanding with no or very little labeled data. Second, we provide some insights on foundational research questions regarding the evaluation of general purpose representation learning methods.

Improving capabilities of image understanding using unlabeled data, especially for pixel-level tasks, opens up a wide range of applications that are beneficial to the society, and which cannot be tackled otherwise. Medical imagery applications suffers from lack of labeled data due to the need of very specialized labelers. Another application, tackling harmful online content—including but not limited to terrorist propaganda, hateful speech, fake news and misinformation—is a huge challenge for governments and businesses. What makes these problems especially difficult is that it is very difficult to obtain clean labeled data for training machine learning models—think of filming a terrorist attack on live video as in the unfortunate Christchurch attack. Self-supervised learning can potentially move the needle in advancing models for detecting extremely rare yet highly impactful incidents. On the other hand, such technologies can be potentially misused for violating privacy and freedom of expression. We acknowledge these risks as being a feature of any amoral technology, and we invite governments, policy makers and all citizens—including the research community—to work hard on striking a balance between those benefits and risks.

Another interesting aspect of our research is highlighting the importance of aligning representation learning methods with the nature of downstream applications. With our method, we show that learning pixel-level representations from unlabeled data we can outperform image-level methods on a variety of dense prediction tasks. Our findings highlight that the research community should go beyond limited test-beds for evaluating generic representation learning techniques. We invite further research on developing comprehensive evaluation protocols for such methods. In fact, we see many research opportunities in the computer vision domain, such as developing a sweep of standardized benchmarks across a variety of geometric and semantic image understanding tasks, and designing methods that can bridge the gap between offline and online performance.

## Funding Disclosure

This work was supported by Element AI.

## Footnotes

[1]We use the official MoCo version 2 for both the baseline and the initialization of VADeR. Training MoCov2 for extra 5M iterations does not give any statistically significant improvement.

[2]We consider `train_aug` for VOC.

[3]We use the evaluation code provided by [67] in: https://github.com/xiaolonw/TimeCycle

[4]Moreover, we consider the default FPN implementation provided on the repository (instead of the one described on this paper) to train VADeR and baselines for these experiments.

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
