[Supplementary Material]

# A   Supplementary Material

## A.1   Implementation Details: feature extraction protocol

**Semantic segmentation.**    The output of each model (baselines or VADeR) follows a $1 \times 1$ conv. layer, $4\times$ upsample and softmax. The final output models the probability of each pixel belonging to a category (21 for PASCAL VOC [17] and 19 for Cityscapes [7]). W train on `train_aug` set of VOC and `train` set of Cityscapes and evaluate on `val` set of each dataset. We use crop size 512 on PASCAL VOC and 768 in Cityscapes and evaluation is done on original image size. Each training sample is generated with random cropping, scaling (by ratio in $[0.5, 2.0]$) and normalization. We train with batch size of 64 for 60 epochs for VOC and batch size 16 and 100 epochs for Cityscapes. In both datasets, the linear layer is trained with SGD (on 4 GPUs) with learning rate 0.005/0.01 (for VOC and Cityscapes), momentum 0.9 and weight decay 0.0001. The learning rate is decayed by a factor of 10 at epochs 42,55 for VOC and 70,90 for Cityscapes.

**Depth Estimation.**    We add a $1 \times 1$ conv. layer on the top of each model to predict a single value at each location. The per-pixel prediction is $4\times$ bilinear-upsampled. We train and evaluate on the official training split of NYU-depth v2 provided by [51]. Both training and evaluation are performed on original image sizes and random horizontal flip is applied to each training sample. We train with batch size of 16 for 30 epochs. The linear layer is trained to minimize the Huber loss between predicted and ground-truth depth values. We use a learning rate of 0.005, momentum 0.9 and weight decay 0.0001. The learning rate is reduced by a factor of 10 at epochs 10,20.

**Video instance segmentation.**    Given the instance mask of an initial frame, the task is to propagate the mask to the rest of the frames. We report results on DAVIS-2017 [59] `val` set. We follow identical evaluation setup of [67, 40] [5]: for frame $t$, we average the the predictions of $[t-7, t]$ to obtain the final propagated map. We use a k-NN propagation schema with $k = 5$.

## A.2   Implementation Details: fine-tunning protocol

**Semantic segmentation.**    We use the same architecture of VADeR (described in 3.2) for all baselines. As above, we transform the 128-dimensional output of each model into a per-pixel probability map with $1 \times 1$ conv, upsampling and softmax. We use the same dataset and preprocessing as described on the feature extraction protocol. We train with batch size of 64 for 60 epochs on VOC and batch size 16 for 100 epochs on Cityscapes. The models are fine-tunned with SGD and learning rate of 0.005/0.01 (for VOC and Cityscapes), momentum 0.9 and weight decay 0.0001.

**Depth Estimation.**    We also use the same VADeR architecture for all baselines. As in the feature extractor protocol, we upsample the predictions to original input size and the model is trained to minimize the Huber loss. We use the same dataset and preprocessing as described on the feature extraction protocol. We train with batch size of 16 for 30 epochs (we reduce the learning rate by 10 on epochs 10,20). The models are fine-tunned with SGD and learning rate of 0.005, momentum 0.9 and weight decay 0.0001.

## A.3   Fine-tuning results with varying number of training samples

Table 5 shows results of the plots in Figure 4. We run experiments with varying number of (labeled) samples. Reported results are averaged over 5 trials.

|          |         | % of data | | | | | |
|----------|---------|------|------|-------|-------|-------|-------|
|          |         | 2    | 5    | 10    | 20    | 50    | 100   |
| VOC      | sup. IN | 37.4 | 48.4 | 53.7  | 59.0  | 69.3  | 73.9  |
| (mIoU)   | MoCo    | 44.7 | 53.3 | 57.2  | 59.3  | 70.1  | 75.3  |
|          | VADeR   | 47.7 | 56.9 | 60.6  | 60.8  | 70.9  | 75.4  |
| NYU-d v2 | sup. IN | 1.288| 1.060| 0.999 | 0.896 | 0.805 | 0.731 |
| (RMSE)   | MoCo    | 1.383| 1.340| 0.897 | 0.804 | 0.707 | 0.640 |
|          | VADeR   | 1.128| 0.951| 0.868 | 0.791 | 0.700 | 0.641 |

Table 5: Results from Figure 4 in tabular format. We show results averaged over 5 trials.

## Footnotes

[5]From https://github.com/xiaolonw/TimeCycle