[Reviews · NeurIPS 2020]

Review 1

Summary and Contributions: This paper extends contrastive learning (and specifically MoCo) to pixels in a straightforward way: apply data augmentations to get two 'views' of an image, and then train a pixel-to-pixel network such that the embeddings of corresponding pixels are more similar than non-corresponding pixels, using a standard InfoNCE loss. Results are competitive across a wide range of benchmarks, outperforming less pixel-aware self-supervised representations for pixel-labeling tasks (segmentation, depth estimation) and generally performing on par for tasks where pixels are less important (e.g. object detection). Strong results are also provided for DAVIS tracking without supervision. === Post rebuttal === Overall, my views remain unchanged after reading the rebuttal and other reviews.

Strengths: Considering the recent success of self-supervised learning with contrastive methods, this is a natural baseline to try when learning representations with pixel labeling as a downstream task. The method is overall a quite straightforward extension of MoCo, and I appreciate the lack of bells and whistles, which makes the results easy to interpret. The results are strong for pixel labeling tasks, without harming performance on other tasks where their baseline, MoCo, performs well. I also find table 3 quite interesting: it seems that invariance to cropping (which loses contextual information that must be recovered) is necessary for keypoints but not for semantic segmentation. This is somewhat mysterious and warrants further investigation.

Weaknesses: The novelty of this paper is relatively small, considering that this is a fairly straightforward update to MoCo to operate on pixel embeddings rather than image embeddings. There seems to be non-trivial engineering in the augmentations, which is not very well ablated: more recent works like SimCLR use a simpler set of augmentations. As this is arguably a 'baseline' due to its simplicity, I would have appreciated more analysis of the important settings in the algorithm. For example, should other pixels in the same image as the input pixel be considered negatives for that pixel? If so, how spatially close can those negatives be before it's a problem? How important is the size of the negative memory bank? Is momentum contrast even needed, or does a SimCLR/CPC-like loss with no momentum encoder work just as well? How important is the MoCo pretraining, and what happens if you remove it? These details matter in practice, and more research is needed on how to set them.

Correctness: Yes

Clarity: Yes

Relation to Prior Work: One minor omission: the paper cites the contribution of Wu et al. "Instance discrimination" as being "a non-parametric version of" Dosovitskiy et al.'s Exemplar Networks. It's worth pointing out that Doersch et al. "Multi-task self-supervised visual learning" was a prior work which proposed and evaluated this same idea. Furthermore, the multi-task paper evaluates several methods on NYUDv2 depth prediction, as this paper does, although it apparently uses an incompatible metric. Ideally the NYUDv2 results here could be made comparable to that earlier work.

Reproducibility: Yes

Additional Feedback:


Review 2

Summary and Contributions: The authors adapt current contrastive representation learning methods to learn dense visual representations. Where previous methods produce a single, global descriptor of an image and maximize its similarity across augmentations, the authors propose to learn a dense grid of descriptors (e.g. a 56x56 grid for a 224x224 image), and maximize the similarity of each location in the image across views. The authors use knowledge of the augmentation process to infer which feature vectors correspond to each other. The authors then use these representations for an array of downstream structured prediction tasks, including semantic segmentation, depth estimation, mask propagation, and keypoint detection. They report strong results on all of these tasks.

Strengths: This work represents the first step away from self-supervised learning of global features, and towards a more task-relevant learning paradigm. The proposed method is simple and elegant, and builds the foundation for a new line of research. However natural the proposed method is, it is to the best of my knowledge completely novel. The empirical benefits of the method are also strong. When evaluating the representation with fixed features, the proposed method outperforms supervised pretraining and MoCo by a large margin. When performing instance mask propagation in this regime, it is also competitive with specialized architectures trained with video data, despite only using a ResNet-50 and training on ImageNet. When the representation is used for fine-tuning the gains are smaller, but they are consistent enough to be convincing. In particular, the method seems to facilitate "geometrical" (e.g. depth estimation and keypoint detection) tasks more than recognition-based ones, and might therefore become more useful as richer tasks are considered.

Weaknesses: The main limitation of the work is that the empirical benefits when fine-tuning are fairly small, considering the self-supervised representation learning method has been tailored for structured prediction tasks. That said, the tasks under consideration are diverse enough that it cannot be said that it has been tailored for any one of them, hence I don't see this as a major limitation.

Correctness: The empirical validation generally appears to be sound. One technical question does come to mind however. Since the method is initialized from MoCo and trained for 50 epochs, it would be good to know what performance MoCo would get, were it to be trained for another 50 epochs. As mentioned previously, since the gains (when fine-tuning) relatively to MoCo are fairly small, it would be reassuring to know that these cannot be attributed to the additional training budget.

Clarity: The paper is very well written and makes for an enjoyable read.

Relation to Prior Work: Prior work is clearly discussed and differentiated from this work.

Reproducibility: Yes

Additional Feedback: **** POST REBUTTAL ***** Thank you for the thorough rebuttal, and for including the additional (low-shot) experiments which address the concern that the experimental gains are incremental. I will maintain my score at 7 (accept). Congratulations on the great work!


Review 3

Summary and Contributions: This paper argues that current contrastive self-learning methods learn image-level representations that are not suitable for visual understanding tasks that require dense representation. Instead, this paper proposes to apply contrastive learning in a pixel-wise manner for unsupervised representation learning, i.e., features from the same pixel should get close, otherwise far in the feature space. The authors claim that the proposed method is more effective that learning at the image level. Downstream tasks have been performed to check the effectiveness of the learned representations.

Strengths: Instead of pushing non-corresponding images apart, e.g., different versions of the same dog image are corresponding images, otherwise not, the method proposes to push non-corresponding pixels apart, e.g., a pair of pixels are considered as correspondence if they are from the same location of an object. Hopefully, this encourages the network to learn more discriminative features. Experiments have demonstrated the effectiveness to some extent but not in a compelling way.

Weaknesses: Very limited novelty. It is more like an intensive data augmentation strategy for contrastive learning in the sense that, one can augment the images in the standard contrastive learning by taking different crops from the same image to get an expanded dataset, where crops from different locations of an image now become non-corresponding images. This gives the equivalent effect that the proposed method is proposing here. A fair comparison should be carried out by applying the aforementioned augmention to constrastive learning that does not use the proposed method here.

Correctness: The claims are not well supported especially on the contributions. The loss function used for contrastive learning is standard.

Clarity: Line 118, why not share parameters between f and g? Line 134, why the burden is only placed on the encoders, are the decoders fixed during training?

Relation to Prior Work: not really. As indicated in the weakness part.

Reproducibility: Yes

Additional Feedback: I would like to keep my rating as weekly accept. Also, I agree with the other reviewers that this is a baseline paper with minor novelty. The suggested baseline of this baseline paper is somewhat mis-intepreted by the authors. And, the suggested baseline is also related to the approximity(radius) proposed by R1. Would like to see it is discussed in the paper.


Review 4

Summary and Contributions: TLDR: simple extension of MOCO with spatial NCE, which gives minor boost in performance on structured vision tasks. This work describes a simple extension of the MOCO framework where instead of contrasting spatially pooled representation, per-pixel (module network stride) "ground-truth" correspondences (obtained by applying random cropping and re-scaling) are used in the noise-contrastive objective function. Comparisons are provided against 3 main initialisation/pre-training schemes: (1) random, (2) supervised imagenet, and (3) MOCO for the same network architecture (ResNet-50). A variety of "structured" tasks, like segmentation, detection, etc. (see "strengths" for full list) are considered, where the proposed method outperforms the baseline methods on most tasks. Note, however, that they start from MOCO pre-trained checkpoint instead of randomly initialized checkpoint.

Strengths: The key strength of this work are strong empirical evaluations on a variety of standard computer vision tasks, namely: (1) segmentation (2) single-image depth (3) object detection (Mask-RCNN style) (4) instance segmentation (5) keypoint regression (6) video instance segmentation (or tracking). The proposed method improves upon MOCO, supervised ImageNet pre-training and random initialization. Another positive aspect is the simplicity of the method.

Weaknesses: 1. A key limitation of this work is that their proposed network VADeR is always initialized with MOCO self-supervised pre-training. While this is benign for practical purposes, it does conflate the two methods, and also means that VADeR is trained for longer etc. Training randomly initialized network with the proposed method will provide crucial empirical evidence, and would only strengthen, not weaken the experiments and claims. 2. The second limitation is that similar works have been proposed, although not in the recent context of memory-bank augmented noise contrastive training. A highly relevant reference is missing: @inproceedings{kanazawa2016warpnet, title={Warpnet: Weakly supervised matching for single-view reconstruction}, author={Kanazawa, Angjoo and Jacobs, David W and Chandraker, Manmohan}, booktitle={Proceedings of the IEEE Conference on Computer Vision and Pattern Recognition}, pages={3253--3261}, year={2016} } This work uses randomly warped geometric "views" for training correspondence networks. 3. It would help to also include results on classification for completeness, e.g., by pooling the learnt features.

Correctness: The claims are correct, and method a simple extension of the MOCO framework. Minor comment: 1. On L101-102 it "viewpoint invariance" is claimed. Note "viewpoint invariance" usually is in the context of 3D transformation of the camera/object, which is not the case here. Please rephrase. "View-agnostic" (as used in section 3.1) is a better term.

Clarity: The writing is generally clear. 1. Confusing notation i L117-123. Clearly introduce f and g as in MOCO. 2. Typo on L260: 'considers'.

Relation to Prior Work: Good discussion of prior work. However, missing 1 reference (see weakness 2).

Reproducibility: Yes

Additional Feedback: 1. Please give examples of "structured, dense predictiction tasks" as alluded to in the abstract and the intro, may be at L31 for completeness and context for the reader. 2. Use em dash "---", instead of en "--" to give more details (as on lines L23, 35, 39, 40, 44 etc.); further em-dashes have no spaces around them (see writing style guides).

[Author Response · NeurIPS 2020]

We thank the reviewers for the detailed and helpful reviews. We particularly acknowledge that reviewers find this work simple (R1, R2, R4), with strong empirical results (R1, R2, R4) and elegant and "completely novel" (R2). Next, we address the main concerns from reviewers.

**Novelty (R1, R3).** Current SSL methods (including MoCo, which we base upon) train only the bottom-up encoder w/o labels. They require pixelwise labels for the top-down decoder, which are initialized from scratch. As pointed by R2, the novelty of the approach is that we learn (unsupervised) pixel-level representations (ie, both the encoder and the decoder) instead of global representations (only the encoder). This allows for better initialization for dense labeling tasks and, to the best of our knowledge, has not been proposed before.

This is fundamentally different than an 'intensive data augmentation', as suggested by R3. By following R3's proposal, we would not be learning any features for the decoder (top-down path). The resulting representation would still encode the entire image and would not be particularly useful to pixel-level tasks. Even if we remove the pooling that proceeds the convolutional layers, the resulting downsampling is too aggressive for pixel-level tasks (factor of 32 in ResNet-50). Therefore, a decoder would still need to be initialized randomly.

**Limited benefit when fine-tuning (R2, R3, R4).** The benefits, however, increase when the amount of labeled data on fine-tuning stages is reduced. The figure on the right shows results (mean/std over 5 runs) when considering only fraction of the total data (2, 5, 10, 20, 50 and 100% of images), for both sem. seg. (VOC) and depth (NYUdepth). This result corroborates current research that show that SSL methods achieve better performance than supervised pre-training when the number of labeled data is limited.

**Comparison with MoCo trained with 50 extra epochs (R2, R4).** Training extra 50 epochs *with the original MoCo augmentations* does not make any statistical difference (on VOC and NYUDv2 fine-tuning) when compared with 200 epochs. This agrees with observations from MoCo-v2 paper (extra training helps much more on linear probing than on fine-tuning). When training the extra 50 epochs *with VADeR augmentations* (that is, same as MoCo, but discarding the pairs that does not share any pixel in common), the performance is slightly worse. This important discussion will be included on the revised version of the paper.

**Training from scratch (R1, R4).** It is not trivial to train randomly initialized networks with top-down path and skip-connections due to interaction between gradient of different paths. We found extremely challenging to train from randomly initialized network due to the very long training time for SSL, the set of hyperparameters and relatively limited resources. Popular methods (U-Net, SharpMask, FPN) start from an encoder initialized with supervised pre-training and random skip-connection/top-down weights. We follow the same approach, but initialize with MoCo weights (unsupervised) instead.

**Missing citations (R1, R4).** We will include the missing references and compare them with this work. (R1:) We will include the metric as in Doersch et al. However, comparison is not apples-to-apples since many training details are different (architecture, loss, etc). (R4:) Compared to Kanazawa et al., we believe the only similarity is the fact that correspondence flow is generated by applying transformation to images. Everything else is different, e.g., the high-level goal, the dataset (imagenet vs. fine-grained CUB), the loss, the evaluated tasks, the learning of features. We believe those works are actually complimentary, and WarpNet (that is initialized with ImageNet pretraining) could benefit from VADeR as other downstream tasks did.

**Non-trivial engineering in augmentations (R1).** We decided to use the same data augmentation as in MoCo-v2 (to facilitate comparison) with one small difference: making sure that at least 32 pixels belong to the two views (so that we can construct a correspondence map between the pixels). In early experiments we tried different minimum number of matching pixels (4, 8, 16, 32) per pair, and did not notice any qualitative difference.

**Settings in the algorithm (R1).** (i) We tried to use different pixels of the same image as negative examples in initial experiments (for proof of concept), but we could not make it work. Using pixels of other images is more natural (no need to try to find the ideal threshold distance) and fits naturally in the context of using a queue for negative samples. (ii) Momentum encoder might not be necessary, but we found easier to scale since SimCLR-like loss requires large batches to run (and VADeR requires even more memory due to the decoder). (iii) We observe very similar behavior w.r.t. to size of memory bank as reported in MoCo, going from 56.8 (with size 4096) mIoU in VOC to 58.3 (with size 65K).

**Why not share params in $f$ and $g$ (L118)? (R3).** Encoders $f$ and $g$ can or cannot share the same parameters. We follow one of the current SSL trends (InstDisc, CMC, PIRL, MoCo, etc.) and use different parameters for $f$ and $g$. The latter is updated with momentum encoder, which allows for effective large number of negative samples.

**Are the decoders fixed during training (L134)? (R3).** No, both the encoder and decoder are trained. We mean "we place all the burden of representation learning on the network parameters (encoder-decoder)" (instead of compatibility function).

**Results on classification for completeness (R4).** We will report results of VADeR on linear probing for completeness on the revised version of the manuscript (although we expect it to be worse than other methods that learn global representations, as this is not our objective).

[Meta-Review · NeurIPS 2020]

Four knowledgeable referees support accept and I accept. We encourage and expect the authors to incorporate the reviewers' suggestions for improving the paper.